# CLaM-TTS: Improving Neural Codec Language Modeling for Zero-shot Text-to-Speech

**Jaehyeon Kim, Keon Lee, Seungjun Chung, Jaewoong Cho**
KRAFTON
{jay.310,keonlee,s.j.chung,jwcho}@krafton.com

## Abstract

With the emergence of neural audio codecs, which encode multiple streams of discrete tokens from audio, large language models have recently gained attention as a promising approach for zero-shot Text-to-Speech (TTS) synthesis. Despite the ongoing rush towards scaling paradigms, audio tokenization ironically amplifies the scalability challenge, stemming from its long sequence length and the complexity of modelling the multiple sequences. To mitigate these issues, we present CLaM-TTS that employs a probabilistic residual vector quantization to (1) achieve superior compression in the token length, and (2) allow a language model to generate multiple tokens at once, thereby eliminating the need for cascaded modeling to handle the number of token streams. Our experimental results demonstrate that CLaM-TTS is better than or comparable to state-of-the-art neural codec-based TTS models regarding naturalness, intelligibility, speaker similarity, and inference speed. In addition, we examine the impact of the pretraining extent of the language models and their text tokenization strategies on performances.

## 1 Introduction

Large language models (LLMs), characterized by a considerable number of model parameters and trained on massive text data, have demonstrated remarkable zero-shot learning capabilities (Brown et al., 2020; Chung et al., 2022; Kaplan et al., 2020). While scaling paradigm affects not only the natural language processing domain but also other fields such as image generation (Ramesh et al., 2021; Saharia et al., 2022), image recognition (Radford et al., 2021), and speech recognition (Baevski et al., 2020b; Radford et al., 2023), significant challenges in their efficient training and inference simultaneously arise. In the realm of image processing, discretizing image representation (Razavi et al., 2019; Ramesh et al., 2021; Esser et al., 2021) has been shown to mitigate these issues by effectively reducing the input length to a manageable size.

Language modeling in the speech domain has become feasible with the emergence of neural audio codecs (Zeghidour et al., 2021; Défossez et al., 2023) that enable high-fidelity audio tokenization. For Text-to-Speech (TTS) synthesis, there have been several attempts to adopt the LLMs for zero-shot TTS, which namely synthesize the diverse speech of any human voice (Zhang et al., 2023; Wang et al., 2023; Kharitonov et al., 2023; Rubenstein et al., 2023). These attempts move away from the previous research direction to train models on curated high-quality recording datasets and produce human-like voices on benchmark datasets (Li et al., 2019; Kim et al., 2021; Tan et al., 2024; Casanova et al., 2022). It is demonstrated that, by training LLMs on tens of thousands of hours of diverse audio data, zero-shot adaptation can be accomplished with just a few seconds of audio input.

Despite the significant advancements in TTS at scale, it still poses challenges to further scale up the models. Neural audio codecs typically generate multiple sequences of audio tokens. For instance, Encodec (Défossez et al., 2023) encodes a 5-second speech into 8 sequences of 375 audio tokens. Several work (Kharitonov et al., 2023; Borsos et al., 2023b) employ the semantic tokens from self-supervised speech representation learning (Chung et al., 2021) as an intermediary between text and audio tokens. Although semantic tokens compress information more concisely than audio tokens, a 5-second speech segment still demands 125 semantic tokens, presenting a hurdle even setting aside the further complexities of audio token modeling from them.

Figure 1: An overview of CLaM-TTS. Training of CLaM-TTS unfolds in two stages: (a) we train a Mel-VAE that encodes a mel-spectrogram to the discrete latent representation from using probabilistic RVQ; (b) using the pre-trained residual vector quantizer from the first-stage, a latent language model, a Gaussian mixture (GM) based latent transformer decoder is trained; The decoder aims to predict latent variables that, when quantized, match with the ground-truth audio tokens.

In this work, we aim to bring the capability of efficient training and inference of large-language models within the TTS domain. To this end, we propose an improved **C**odec **La**nguage **M**odel-based **TTS** (**CLaM-TTS**) system that encodes speech into multiple token sequences similar to existing methods but in a more concise way. With CLaM-TTS, all multiple tokens at each timestep in these sequences are generated through a single autoregressive step of a language model, eliminating the need for iterative generative modeling along the number of sequences. The core of our method lies in the probabilistic discrete representation learning, ensuring that all discrete latent codes participate in the training process, resulting in a high-quality autoencoder for speech. Furthermore, we provide a principled framework enabling a latent language model to efficiently generate a stack of tokens at once; the latent language model produces a continuous latent audio representation and converts it to a discrete representation with the proposed probabilistic quantization method. We scale up the training dataset to 100K hours. Our experimental findings indicate that CLaM-TTS either surpasses or is on par with leading zero-shot neural codec-based TTS models in aspects such as naturalness, intelligibility, speaker similarity, and inference speed. Furthermore, we investigate how the depth of pretraining in the language models and their methods of text tokenization influence TTS outcomes. Our generated samples are available on our demo page[1].

## 2 RELATED WORK

**Neural audio codec**   The neural discrete representation learning within a variational autoencoder (VAE) framework, called the vector-quantized VAE (VQ-VAE), has been proven effective in encoding raw-waveforms into discrete tokens (Baevski et al., 2020a), employed as a speech codec (Oord et al., 2017; Gârbacea et al., 2019). Similar to VQ-VAE, the neural audio codec methods usually use a framework that jointly trains an encoder, a decoder, and a quantizer (Li et al.; Zeghidour et al., 2021; Jiang et al., 2022; Jayashankar et al., 2022; Défossez et al., 2023; Kumar et al., 2023; Wu et al., 2023). Zeghidour et al. (2021) pioneers using residual vector quantization (RVQ) (Gray, 1984; Vasuki & Vanathi, 2006; Lee et al., 2022) in a neural audio codec model. It operates efficiently on clean and noisy speech and music, even at low bitrates. EnCodec (Défossez et al., 2023) employs a similar model structure with improved training efficiency and stability to achieve a downsampling rate of 320 for input waveforms. Kumar et al. (2023) identify the issue of codebook under-utilization in EnCodec and improve the codebook utilization with the techniques introduced in Yu et al. (2021) resulting in state-of-the-art performance as a neural audio codec.

---

[1]https://clam-tts.github.io

Building on these advancements, we focus more on the discrete representation learning of speech rather than general audio and optimize the compression level to be suitable for the TTS task. In other words, we compress mel-spectrograms rather than raw waveforms, delegating the task of converting mel-spectrograms back into raw waveforms to standard vocoders.

**Large-scale TTS** AudioLM (Borsos et al., 2023a) is a language model directly trained on audio tokens. In AudioLM, semantic tokens are first generated. These tokens originate from self-supervised discrete speech representation methods (Hsu et al., 2021; Chung et al., 2021) that have previously been utilized for speech resynthesis or generation without text (Lakhotia et al., 2021; Polyak et al., 2021; Nguyen et al., 2023). Following this, the model produces acoustic tokens of neural audio codes given the semantic tokens. Wang et al. (2023) propose the first neural codec language model, Vall-E, for text-to-speech that utilizes a pre-trained neural audio codec, EnCodec (Défossez et al., 2023). In a different approach, following AudioLM, text-to-speech has been realized by applying language modeling to generate the semantic tokens from text, as demonstrated by Kharitonov et al. (2023). A shared characteristic among these neural codec language models is their two-stage pipeline; they autoregressively generate coarse-grained audio tokens and decode them into fine-grained representations. Recent work in music generation hints at an efficient way to eliminate the second-stage modeling by interleaving audio tokens in a delayed pattern (Copet et al., 2023), but its application in TTS remains unexplored.

Given the complexities in modeling long audio sequences, several studies have incorporated phonemes and durations to alleviate the need for speech synthesizers to predict speech rates (Shen et al., 2023; Le et al., 2023; Jiang et al., 2023). Some work shows that non-autoregressive generative models, such as a diffusion model and flow-matching (Ho et al., 2020; Lipman et al., 2023), can produce diverse and natural-sounding speech with large-scale training. A hybrid method is utilized in another approach, employing non-autoregressive architecture except prosody modeling (Jiang et al., 2023). This method aligns with previous work that applies VQ-VAEs to capture fine-grained speech features so that the prosody is controllable by them (Sun et al., 2020; Ren et al., 2022).

To address the challenges associated with neural codec language models while not relying on the phoneme and its duration that requires significant domain expertise, we design a language model that generates from coarse to fine-grained tokens without needing a two-stage pipeline. Our approach is similar to recent work that utilizes pre-trained language models, Spectron (Nachmani et al., 2023) and SoundStorm (Borsos et al., 2023b). While Spectron employs pre-trained transformer decoders to directly model the mel-spectrogram and then fine-tunes it, our method preserves the pre-trained text encoder and decodes speech tokens that are shorter than the mel-spectrogram using a latent transformer decoder. SoundStorm freezes a pre-trained text encoder similar to ours, but it generates semantic tokens and subsequently decodes acoustic tokens using an iterative generative model.

## 3 PRELIMINARIES

Building upon the approach proposed by Wang et al. (2023), we explore zero-shot TTS through the lens of neural codec language modeling task. We consider a setting that includes two types of data: (i) text data and (ii) mel-spectrogram representation of its corresponding speech data, denoted by $x$ and $y$, respectively. We model a sequence of $T$ discrete codes $c_{1:T} := \{c_1, \ldots, c_T\}$ from latent representations $z_{1:T}$ of a mel-spectrogram $y$, using the framework of a Variational Autoencoder (VAE) with Residual Vector Quantization (RVQ). Here, $c_t$ represents the $D$-depth of quantized, discrete codes. We interchangeably use $c_{t,1:D}$ with $c_t$. Subsequently, a neural language model $p_\theta(c_{1:T}|x)$ is employed, aiming to predict $c_{1:T}$ from the text transcript $x$. During the inference phase, the language model generates $c_{1:T}$ for a given text $x$, which is subsequently transformed into speech through the VAE decoder and a pre-trained vocoder.

### 3.1 RESIDUAL-QUANTIZED VARIATIONAL AUTOENCODER (RQ-VAE)

An RQ-VAE (Lee et al., 2022) is a neural network architecture representing data as discrete codes using residual vector quantization. It comprises of three components: (1) an encoder parameterized by $\phi$ that maps data $y$ into a sequence of latent representations $z_{1:T}$; (2) a residual vector quantizer $\mathsf{RQ}_\psi(\cdot)$, converting the latent vector $z_t$ at each time $t$ into the discrete code representation $c_{t,1:D} =$

$\mathrm{RQ}_\psi(\boldsymbol{z}_t)$, or the corresponding quantized embedding $\hat{\boldsymbol{z}}_t$; and (3) a decoder parameterized by $\omega$ that reconstructs the data $\hat{\boldsymbol{y}}$ from a sequence of the quantized latent representations $\hat{\boldsymbol{z}}_{1:T}$.

Here $\boldsymbol{c}_{t,1:D}$ represents the set $\{c_{t,1}, \ldots, c_{t,D}\}$ with $D$ indicating the total depth of the quantizer. The latent representation from the encoder is quantized through the multi-stage nearest-neighbour lookup over the codebook embeddings, of which the vocab size is $V$. The process is defined as finding the optimal code from the codebook, which minimizes the residual error at each depth $d$:

$$c_{t,d} = \underset{c' \in \{1,\ldots,V\}}{\arg\min} \ \|\boldsymbol{r}_{t,d-1} - e_\psi(c';d)\|^2, \quad \boldsymbol{r}_{t,d} = \boldsymbol{r}_{t,d-1} - e_\psi(c_{t,d};d) \quad \text{for all } d \in [1,D], \quad (1)$$

where $\boldsymbol{r}_{t,0} = \boldsymbol{z}_t$ and $e_\psi(c;d)$ corresponds to the $c$-th embedding vector in the codebook at depth $d$. The sum of embeddings $\sum_{d=1}^{D} e_\psi(\boldsymbol{c}_{t,d};d)$ becomes the quantized latent representation $\hat{\boldsymbol{z}}_t$, which is converted back to the input space through the decoder. The codebook embeddings are updated with the clustered latents by the exponential moving average updates (Razavi et al., 2019).

## 3.2 Mean-field Variational Inference

Consider a latent variable model characterized by a joint distribution $p_\psi(\boldsymbol{z}_t, \boldsymbol{c}_{t,1:D})$ parameterized by $\psi$. Here, $\boldsymbol{z}_t$ denotes an observed random variable, $\boldsymbol{c}_{t,1:D}$ indicates a set of latent random variables $\{\boldsymbol{c}_{t,1}, \ldots, \boldsymbol{c}_{t,D}\}$. In this model, variational inference is a method to approximate the intractable distribution $p_\psi(\boldsymbol{c}_{t,1:D}|\boldsymbol{z}_t)$ by solving an optimization problem with respect to parameters of approximate distribution $q(\boldsymbol{c}_{t,1:D}|\boldsymbol{z}_t)$. We can derive a lower bound on the marginal log-likelihood $p_\psi(\boldsymbol{z}_t)$, known as the evidence lower bound (ELBO) (Blei et al., 2017):

$$\log p_\psi(\boldsymbol{z}_t) = \log \sum_{\boldsymbol{c}_{t,1:D}} p_\psi(\boldsymbol{z}_t|\boldsymbol{c}_{t,1:D}) p(\boldsymbol{c}_{t,1:D}) \geq \mathbb{E}_{q(\boldsymbol{c}_{t,1:D}|\boldsymbol{z}_t)} \left[ \log \frac{p_\psi(\boldsymbol{z}_t|\boldsymbol{c}_{t,1:D}) p(\boldsymbol{c}_{t,1:D})}{q(\boldsymbol{c}_{t,1:D}|\boldsymbol{z}_t)} \right].$$

Mean-field variational inference (Koller & Friedman, 2009; Blei et al., 2017) is a specific approach of variational inference that assumes the independence among the latent variables conditioned on the observed variable: $q(\boldsymbol{c}_{t,1:D}|\boldsymbol{z}_t) = \prod_{d=1}^{D} q(\boldsymbol{c}_{t,d}|\boldsymbol{z}_t)$. We can show that each optimal variational posterior distribution $q^*(\boldsymbol{c}_{t,d}|\boldsymbol{z}_t)$, which maximizes the ELBO, satisfies:

$$q^*(\boldsymbol{c}_{t,d}|\boldsymbol{z}_t) \propto \exp\left(\mathbb{E}_{q(\boldsymbol{c}_{t,-d}|\boldsymbol{z}_t)}[\log p_\psi(\boldsymbol{z}_t|\boldsymbol{z}_{t,d}, \boldsymbol{z}_{t,-d}) p(\boldsymbol{c}_{t,d}, \boldsymbol{c}_{t,-d})]\right), \quad (2)$$

where $\boldsymbol{c}_{t,-d}$ denotes the latent variables of all depth $\boldsymbol{c}_{t,1:D}$ except $\boldsymbol{c}_{t,d}$. An iterative coordinate ascent algorithm based on Eq. 2 can be used to update the distribution $q$ (Bishop & Nasrabadi, 2006), and the complexity of the algorithm mainly lies on the computation of the expectation over $q(\boldsymbol{c}_{t,-d}|\boldsymbol{z}_t)$.

# 4 Method

## 4.1 Mel-VAE

We aim to develop a neural codec that can generate discrete speech codes within a short sequence length to make speech audios suitable for language model utilization. To achieve this, we employ a RQ-VAE that compresses mel-spectrograms of speech audios (see Fig. 1a). We introduce a variational inference based method for learning residual codewords to address the codeword collapse issue found in conventional vector quantization methods (Kaiser et al., 2018; Roy et al., 2018; Zeghidour et al., 2021; Kumar et al., 2023).

We illustrate Mel-VAE similar to RQ-VAE following most of notations from Sec. 3.1. The encoder maps a mel-spectrogram $\boldsymbol{y}$ into a sequence of latent representations $\boldsymbol{z}_{1:T}$, and a residual vector quantizer $\mathrm{RQ}_\psi(\cdot)$, converting the latent vector $\boldsymbol{z}_t$ at each time $t$ into the discrete code representation $\boldsymbol{c}_t$, or its corresponding quantized embedding $\hat{\boldsymbol{z}}_t = \sum_{d=1}^{D} e_\psi(\boldsymbol{c}_{t,d};d)$. The decoder reconstructs the mel-spectrogram $\hat{\boldsymbol{y}}$ from a sequence of quantized latent representations $\hat{\boldsymbol{z}}_{1:T}$.

With the assumptions that $q(\boldsymbol{c}_{t,1:D}|\boldsymbol{z}_t) = \prod_{d=1}^{D} q(\boldsymbol{c}_{t,d}|\boldsymbol{z}_t)$ and $p(\boldsymbol{c}_{t,d}, \boldsymbol{c}_{t,-d})$ is uniformly distributed, mean-field variational inference yields the condition of such distribution as the following (see Eq. 2):

$$q^*(\boldsymbol{c}_{t,d}|\boldsymbol{z}_t) \propto \exp(\mathbb{E}_{q(\boldsymbol{c}_{t,-d}|\boldsymbol{z}_t)}[\log p_\psi(\boldsymbol{z}_t|\boldsymbol{c}_{t,d}, \boldsymbol{c}_{t,-d})]), \quad (3)$$

where the latents follows a normal distribution: $p_\psi(z_t|c_t) = \mathcal{N}(z_t; \sum_d e_\psi(c_{t,d}; d), \sigma_\psi^2 I)$.

However, the mutual interdependence of codes at every depth in the latter equation makes it difficult to solve it without an iterative approach. Instead of using an iterative coordinate update approach, we approximate $\mathbb{E}_{q(c_{t,-d}|z_t)}[\log p_\psi(z_t|c_{t,d}, c_{t,-d})]$ pointwisely as $\log p_\psi(z_t|c_{t,d}, c_{t,-d}^*)$ for all $d$, where $c_{t,1:D}^* = \mathsf{RQ}_\psi(z_t)$. The posterior then has a form: $q^*(c_{t,d}|z_t) \propto p_\psi(z_t|c_{t,d}, c_{t,-d}^*)$.

We independently optimize the codebook embeddings at each depth $d$, in a variational inference framework:

$$\mathcal{L}(\psi_d; z_t, c_{t,-d}^*) = \mathbb{E}_{q^*(c_{t,d}|z_t)}\left[-\log p_\psi(z_t|c_{t,d}, c_{t,-d}^*)\right], \tag{4}$$

$$\mathcal{L}(\psi; z_t, c_{t,1:D}^*) = \sum_{d=1}^{D} \mathcal{L}(\psi_d; z_t, c_{t,-d}^*). \tag{5}$$

The other modules of Mel-VAE, including the encoder parameters $\phi$ and the decoder parameters $\omega$, are trained with commitment loss, reconstruction loss, and adversarial losses.

$$\mathcal{L}(\omega, \phi; y, c_{1:D}) = \lambda_r|y - \hat{y}| + \lambda_c\|z - \sum_d e_\psi(c_{t,d}; d)\|^2 + \lambda_a\mathcal{L}_{adv}, \tag{6}$$

where $\lambda_r$, $\lambda_c$, and $\lambda_a$ corresponds to coefficients of the reconstruction loss, commitment loss, and adversarial losses, respectively. For adversarial training, we adopt the multi-length discriminator (Chen et al., 2020) that distinguishes mel-spectrograms at different lengths and a modified multi-resolution spectrogram discriminator (Lee et al., 2023) that accepts mel-spectrogram rather than linear spectrogram. We combine the least squares GAN objective (Mao et al., 2017) and the $L1$ feature matching loss (Kumar et al., 2023) into a loss function denoted by $\mathcal{L}_{adv}$.

## 4.2 LATENT LANGUAGE MODELING

We propose a conditional speech code language model given text $x$ aimed at enhancing the efficiency of the model. This improvement stems from the insight that speech codes are generated through vector quantization. Our approach leverages this by predicting the vector itself, which can then be converted into multiple tokens at each layer via residual vector quantization. This is a departure from previous methods that predict speech code tokens sequentially.

Specifically, rather than directly predicting tokens from text, we consider a continuous latent representation $z_t$ of speech that can be converted into a speech code using residual vector quantization:

$$p_\theta(c_{1:T}|x) = \prod_{t=1}^{T} p_\theta(c_t|x, c_{<t}) = \prod_{t=1}^{T} \int p_\theta(c_t, z_t|x, c_{<t})dz_t = \prod_{t=1}^{T} \int p_\theta(z_t|x, c_{<t})p_\psi(c_t|z_t)dz_t,$$

where $c_{<t}$ indicate $c_{1:t-1}$, and we employ $p_\psi(c_t|z_t)$, the probabilistic quantizer distribution learned together with the Mel-VAE model (see Sec. 4.1), as a replacement of $p_\theta(c_t|z_t, x, c_{<t})$. Here, we define the conditional distribution $p_\theta(z_t|x, c_{<t})$ as a Gaussian mixture model:

$$p_\theta(z_t|x, c_{<t}) = \sum_{k=1}^{K} p_\theta(k|x, c_{<t})\mathcal{N}(z_t; \mu_\theta(k, x, c_{<t}), \sigma_\psi^2 I).$$

In this model, we can derive the following variational lower bound on the log-likelihood:

$$\log p_\theta(c|x) \geq \sum_{t=1}^{T} \mathbb{E}_{q(k|x,c_{\leq t})}\left[-D_{KL}(p_\psi(z_t|c_t)||p_\theta(z_t|x, c_{<t}, k)) + \log p_\theta(k|x, c_{<t}) + \mathcal{B}(\psi, c_t)\right]$$

$$= -\mathcal{L}_{\mathsf{VB}}(\theta) + \mathcal{B}(\psi, c_t),$$

for any $q(k|x, c_{\leq t})$. The derivation of the lower bound and the definition of $\mathcal{B}(\psi, c_t)$ are provided in Appendix A. Here, we set $q(k|x, c_{\leq t}) \propto \exp(-D_{KL}(p_\psi(z_t|c_t)||p_\theta(z_t|x, c_{<t}, k)))$.

With the second loss $\mathcal{L}_{\mathsf{EOS}}(\theta)$, which is associated with training a binary classifier to identify the end of speech (EOS), the total loss for training the latent language model is the sum of the two losses above: $\mathcal{L}(\theta) = \mathcal{L}_{\mathsf{VB}}(\theta) + \mathcal{L}_{\mathsf{EOS}}(\theta)$.

As shown in Fig. 1, we implement an autoregressive latent model that yields three distinctive outputs: the mixture weights and the means of the Gaussian mixture distribution as well as the probability of concluding the generation. Specifically, it incorporates a transformer decoder followed three parallel modules, comprising (1) a prediction layer with softmax activation for $p_\theta(k|x, c_{<t})$; (2) a prediction layer for $\mu_\theta(k, x, c_{<t})$; (3) a binary classifier layer for EOS prediction. Additionally, we use the pre-trained quantizer $\mathsf{RQ}_\psi(\cdot)$ of Mel-VAE.

### 4.3 MODEL ARCHITECTURE AND INFERENCE

**Model Architecture** For the Mel-VAE, we adopt a causal 1d convolutional U-Net, a variant of the model used in Ho et al. (2020). We remove the skip connections and attention layers and append 1-d ConvNeXt (Liu et al., 2022) blocks used in Siuzdak (2023) to the final layer of the decoder. We employ 32-stage residual vector quantization with a codebook size of 1,024 for each depth. For the text-to-code latent language model, we adopt a transformer-based encoder-decoder LM, especially a pre-trained ByT5-large[2] (Xue et al., 2021a) similar to Borsos et al. (2023b). We keep the text encoder frozen. Please refer to Appendix C for more detailed model configuration.

**Inference** The text-to-code generation unfolds in three steps: (1) at time step $t$, we randomly select $k$ from the distribution $p_\theta(k|\boldsymbol{x}, \boldsymbol{c}_{<t})$; (2) following this, randomly sample the latent vector $\boldsymbol{z}_t$ from $p_\theta(\boldsymbol{z}_t|\boldsymbol{x}, \boldsymbol{c}_{<t}, k)$. Consequently, at time step $t$, the discrete code is obtained through the learned quantizer, $\boldsymbol{c}_t = \mathsf{RQ}_\psi(\boldsymbol{z}_t)$; (3) if the probability of EOS exceeeds 0.5, conclude the generation, or proceed to step, otherwise. Subsequently, the generated codes are decoded to mel-spectrograms using the decoder of Mel-VAE, then converted to raw-waveforms through an off-the-shelf pre-trained vocoder, BigVGAN (Lee et al., 2023).

## 5 EXPERIMENTAL SETUP

**Training Dataset** We employ 100K hours of over 12K distinct speakers' speech-transcript dataset spanning 11 languages: English, Korean, Chinese, Japanese, German, Dutch, French, Spanish, Italian, Portuguese, and Polish. We train two models: (1) CLaM-en: an English-only model on 55K-hour English dataset and (2) CLaM-multi: a multilingual model trained on 11-language dataset. We provide details of dataset for each language in Appendix B.1, and data pre-processing in Appendix B.2 and B.3.

**Training** (1) Mel-VAE: We train the model on 4 NVIDIA A100 40GB GPUs for around 2M steps. Each GPU processes a size one minibatch containing concatenated mel-spectrograms of several audios. We trim the trailing end to have it precisely 32,768 frames long. We use Adam optimizer (Kingma & Ba, 2015) with a constant learning rate of 0.0002 throughout the training. (2) Text-to-code: We train only the decoder and use a learned codebook from Mel-VAE. The model is trained on 4 NVIDIA A100 40GB GPUs for around 4M steps with dynamic batching while keeping a maximum code size of 2,560. We use AdamW optimizer (Loshchilov & Hutter, 2019), and the learning rate is fixed to 0.0002 throughout the training. Throughout all our experiments, during the model inference, we sample $k$ using top-$p$ sampling (Holtzman et al., 2020) with 0.5 and $z$ is sampled with temperature (Kingma & Dhariwal, 2018) of 2.6, which matches the empirical standard deviation in our validation dataset.

**Baselines** We compare the proposed model with the following four baselines: (1) YourTTS (Casanova et al., 2022), a zero-shot TTS built on VITS (Kim et al., 2021) which is flow-based end-to-end TTS (representing **Conventional TTS**), (2) Vall-E (Wang et al., 2023) and (3) SPEAR-TTS (Kharitonov et al., 2023) (representing **Neural Codec LM**), and (4) VoiceBox (Le et al., 2023), a flow-matching-based TTS model trained on large-scale training data (representing **Non-Autoregressive Model with Phoneme Input and Duration**).

**Metrics** (1) **Intelligibility and Robustness**: We measure these attributes by character error rate (CER) and word error rate (WER) of the synthesized transcription from generated speech concerning the input text. We follow the procedure in Wang et al. (2023). In English-only Evaluation, we synthesize the transcription by using the automatic speech recognition (ASR) model, the CTC-based HuBERT-Large[3] (Hsu et al., 2021) model pre-trained on LibriLight (Kahn et al., 2020) and then fine-tuned on LibriSpeech (Panayotov et al., 2015). In the Multilingual Evaluation, we use OpenAI's Whisper (Radford et al., 2023) model[4]. We adopt NVIDIA's NeMo-text-processing[5] (Zhang et al., 2021; Bakhturina et al., 2022) for text normalization; (2) **Speak Similarity**: We assess the speaker similarity of two separate speech audio clips by following the same procedure outlined in Wang et al. (2023). We employ WavLM-TDCNN[6] (Chen et al., 2022) which outputs the embed-

---

[2] https://huggingface.co/google/byt5-large

[3] https://huggingface.co/facebook/hubert-large-ls960-ft

[4] https://github.com/openai/whisper/blob/main/model-card.md: "large-v2"

[5] https://github.com/NVIDIA/NeMo-text-processing

[6] https://github.com/microsoft/UniSpeech/tree/main/downstreams/speaker_verification

ding vector representing the speaker's voice attribute. We measure the cosine similarity between the two embedding vectors to get a score in $[-1, 1]$, where a higher score indicates a higher speaker similarity of the audios. We borrow the definition of SIM-o and SIM-r from Le et al. (2023). SIM-o measures the similarity between the generated and the original target speeches, while SIM-r measures the similarity concerning the target speech reconstructed from the original speech by Mel-VAE and the pre-trained vocoder; (3) **Subjective Speech Quality**: We measure the quality of the generated speech from human evaluations via three types of Mean Opinion Score (MOS) (Ribeiro et al., 2011): i) Quality MOS (QMOS) for an overall audio assessment, ii) Similarity MOS (SMOS) to measure speaker similarity between the prompt and the generated speech, and iii) Comparative MOS (CMOS) to compare our model with available baselines. Detailed settings of subjective tests are described in Sec. B.5.

**Tasks**    We measure the performances of the proposed model under two different tasks: 1) *continuation*: Given a text and corresponding initial 3 seconds of the Ground Truth speech as a prompt, the task is to seamlessly synthesize the subsequent portion of the speech, and 2) *cross-sentence*: The model is given a text, a 3-second speech segment, and its corresponding transcript (the transcript is different from the text). The task is to synthesize a speech reading the text in the style of the provided 3-second speech. We include our samples across the tasks discussed above, covering speaker diversity, text prompting, and other aspects, on our demo page.

# 6    EXPERIMENTAL RESULTS

## 6.1    ENGLISH-ONLY EVALUATIONS

**Evaluation Methodology**    We evaluate performances of CLaM-en across *continuation* and *cross-sentence* tasks. Following the evaluation setting in Wang et al. (2023), we employ a subset of the LibriSpeech test-clean dataset. This subset comprises speech clips ranging from 4 to 10 seconds, each with a corresponding transcript. Note that YourTTS has official checkpoints, Vall-E has an unofficial checkpoint[7], and others do not have checkpoints. We use checkpoints of YourTTS and Vall-E for evaluations. We compare the other baselines with ours via the performances reported in their papers (Wang et al., 2023; Kharitonov et al., 2023; Le et al., 2023). Since SPEAR-TTS and VoiceBox also evaluate using the same approach with Vall-E, they can be directly compared with our model as well. Details of evaluation are provided in Appendix B.4.

**Analysis**    Tab. 1 and 2 show the results of *continuation* and *cross-sentence* task, respectively. Ours offers great performances for all measures, ranking either first or second, except SIM-r in cross-sentence task.

It is worth noting that VoiceBox (Le et al., 2023), a phoneme-based duration model, shows better performances than ours. However, it requires both phoneme and duration for speech synthesis, whereas our model directly employs a pretrained language model. This allows for the seamless integration of LMs, which are trained across a broad spectrum of texts and tasks, enabling a plug-and-play methodology. Experimental results of training several T5 variants is shown in Appendix D.2, illustrating the trade-off between leveraging the inherent capacity of LMs and ensuring robustness. We also compare the end-to-end inference time for a 10-second utterance. Our method is faster than the generation speed of Vall-E reported in Le et al. (2023). While ours is faster than VoiceBox with 64 decoding steps, VoiceBox can use fewer iterations of decoding steps. Tab. 3 presents the subjective audio evaluations. CLaM-en significantly outperforms the baseline, YourTTS, in quality and intelligibility, as indicated by QMOS. Our adherence to the prompt audio surpasses that of the baseline, as measured by the SMOS. The comparative scores (CMOS) highlight CLaM-en's proximity to the Ground Truth regarding naturalness, clarity, and comprehensibility. Overall, CLaM-en's generated speech naturalness, quality, intelligibility, and similarity exceed the baseline.

## 6.2    MULTILINGUAL EVALUATIONS

We evaluate our model, CLaM-multi trained on the multilingual dataset. On the test set, we measure WER, CER, and SIM-o which are defined in Sec. 5. Here, we only consider *continuation* task in this experiment since we cannot get full alignmnets between audio and text for all languages and datasets. Tab. 4 shows the partial results of the multilingual *continuation* task. We sample a hundred

---

[7]https://github.com/lifeiteng/vall-e

Table 1: Performances for the English-only *continuation* task. The boldface indicates the best result, the underline denotes the second best, and the asterisk denotes the score reported in the baseline paper. Ours offers great performances for all measures, ranking either first or second. The inference time indicates the generation time of 10s speech.

| Model | WER ↓ | CER ↓ | SIM-o ↑ | SIM-r ↑ | Inference Time ↓ |
|---|---|---|---|---|---|
| Ground Truth | 2.2* | 0.61* | 0.754* | 0.754* | n/a |
| YourTTS (Casanova et al., 2022) | 7.57 | 3.06 | 0.3928 | - | - |
| Vall-E (Wang et al., 2023) | 3.8* | - | 0.452* | 0.508* | ∼6.2s* |
| Vall-E (unofficial) | 3.81 | 1.58 | 0.2875 | 0.3433 | - |
| Voicebox (Le et al., 2023) | **2.0*** | - | **0.593*** | **0.616*** | ∼6.4s* (64 NFE) |
| CLaM-en | 2.36 | **0.79** | 0.4767 | 0.5128 | **4.15s** |

Table 2: Performances for the English-only *cross-sentence* task.

| Model | WER↓ | CER↓ | SIM-o↑ | SIM-r↑ |
|---|---|---|---|---|
| YourTTS (Casanova et al., 2022) | 7.92 (7.7*) | 3.18 | 0.3755 (0.337*) | - |
| Vall-E (Wang et al., 2023) | 5.9* | - | - | 0.580* |
| Vall-E (unofficial) | 7.63 | 3.65 | 0.3031 | 0.3700 |
| SPEAR-TTS (Kharitonov et al., 2023) | - | **1.92*** | - | 0.560* |
| Voicebox (Le et al., 2023) | **1.9*** | - | **0.662*** | **0.681*** |
| CLaM-en | 5.11 | 2.87 | 0.4951 | 0.5382 |

random samples from the test set of each dataset, ranging from 4 to 20 seconds, and average the scores of three trials. Refer to Tab. 10 for evaluation on other languages and datasets.

## 6.3 ABLATION STUDY

**Effectiveness of Proposed RVQ**  To demonstrate the effect of the proposed RVQ on Mel-VAE, we conduct an ablation study by assessing speech reconstruction capability. We train two Mel-VAEs: one with the proposed RVQ and the other with the baseline RVQ (Défossez et al., 2023). We train both for 500k steps on the same training dataset described in Sec. 5. The generated speech is compared to the Ground Truth speech using two metrics: Perceptual Evaluation of Speech Quality (PESQ) (Rix et al., 2001) and Virtual Speech Quality Objective Listener (ViSQOL) (Chinen et al., 2020) in speech mode. For evaluation, we randomly select 1,800 samples from the test set of each dataset, proportional to the size of each dataset, each being at least 3 seconds long. The scores of these samples are then averaged for comparison. Tab. 5a shows that ours is more effective than the baseline RVQ. See Fig. 2 to verify the superior codebook usage of our approach. We also compare the fully trained Mel-VAE with Encodec at 6K bitrates (Défossez et al., 2023), which is widely employed in neural codec language models (Wang et al., 2023; Zhang et al., 2023). Tab. 5b confirms that ours outperforms Encodec in speech reconstruction performance across both measures.

**Comparision of Pre-trained LM and Input Variants**  Our language model is based on T5 (Raffel et al., 2020). We conduct an ablation studty to compare T5, its variants and a phoneme encoder of comparable size. The results indicate that ByT5 surpasses other T5 variants with the sole exception of the phoneme model. This suggests that: 1) the more the pretraining phase is leveraged, the greater the potential increase in TTS performance, and 2) in moderate-sized language modeling, phonemes remain an effective input representation. For the experimental results and a comprehensive analysis, refer to Appendix D.2.

In addition to the ablation studies presented, we have conducted further experiments detailed in the appendix, which explore the effects of codeword emitting rate on speech codec quality and language modeling as well as the scale of training data on model efficacy. For comprehensive results and discussions, refer to Appendix D.3 and D.4.

## 7 DISCUSSION

**Choice of Codeword Rate**  Our approach enjoys a 10Hz codeword rate for efficient modeling. We set the codeword rate following the average phoneme rate in English speech (Roach, 2009) since phoneme is the minimum spoken unit. Nevertheless, we conjecture this may have to be ad-

Table 3: Human evaluations with 40 LibriSpeech test-clean speakers show CLaM-en's speech generation surpasses the baseline in quality, intelligibility, similarity, and naturalness, nearing Ground Truth. QMOS and SMOS scores include a 95% confidence interval.

| Model | QMOS | SMOS | CMOS (vs. CLaM-en) |
|---|---|---|---|
| YourTTS (Casanova et al., 2022) | $2.39_{\pm 0.19}$ | $2.32_{\pm 0.21}$ | -1.68 |
| CLaM-en | $3.87_{\pm 0.12}$ | $3.49_{\pm 0.14}$ | 0.00 |
| Ground Truth | $4.45_{\pm 0.09}$ | $4.18_{\pm 0.15}$ | +0.63 |

Table 4: Performances of CLaM-multi for the multilingual *continuation* task.

| Language / Dataset | WER↓ | CER↓ | SIM-o↑ |
|---|---|---|---|
| English / MLS English | 8.71 | 5.19 | 0.4000 |
| English (HuBERT) / MLS English | 7.71 | 3.19 | 0.4000 |
| German / MLS German | 9.63 | 4.11 | 0.4219 |
| Dutch / MLS Dutch | 12.25 | 4.97 | 0.5983 |
| French / MLS French | 10.29 | 4.08 | 0.5671 |
| Spanish / MLS Spanish | 4.02 | 1.91 | 0.5292 |
| Italian / MLS Italian | 19.70 | 5.19 | 0.5459 |
| Portuguese / MLS Portuguese | 9.66 | 3.72 | 0.5658 |
| Polish / MLS Polish | 14.70 | 5.34 | 0.5519 |

Table 5: Effectiveness of our proposed RVQ. The results show that ours outperforms the conventional RVQ in (a) and Encodec in (b) across both measures, even with a higher compression rate.

(a)

| Model | PESQ↑ | ViSQOL↑ |
|---|---|---|
| ours + BigVGAN | **2.63** | **4.48** |
| RVQ + BigVGAN | 2.54 | 4.44 |

(b)

| Model | PESQ↑ | ViSQOL↑ |
|---|---|---|
| ours + BigVGAN | **2.95** | **4.66** |
| Encodec | 2.59 | 4.26 |

justed depending on the language or speaker. A more compressed codeword rate, for example, 5Hz, might lead to more significant information loss than their efficiency. There exists an efficiency-performance tradeoff for rates above 10Hz, which can be optimized as needed.

**Robustness**  We have noticed some words can be muddled, omitted, or repeated, which predominantly stems from autoregressive modeling. We will address it by employing non-autoregressive architecture or improving the attention mechanism in future work.

**Expressiveness**  100K hours of training data may not ensure a complete representation of all voice types, especially accentuated ones. Our datasets predominantly capture audiobook reading styles, leading to limited diversity in speaking styles. We believe that increasing the model and data size can significantly tackle the expressiveness challenges in zero-shot TTS.

**Instruction Prompting**  We suggest various ways to use the full knowledge of the language model. One can incorporate speaker metadata into each transcript to perform various intriguing tasks. Such tasks might include synthesizing speech or even conversations characterized by specific genders, voice ages, or accents. We leave the other tasks for future work.

## 8    CONCLUSION

We introduce CLaM-TTS, which leverages mean-field variational inference based probabilistic residual vector quantization (1) achieving significant compression in token length, and (2) allowing a latent language model to generate multiple tokens at once, thereby eliminating the need for cascaded modeling to handle the number of token streams. We scale up the training dataset to 100K hours. We empirically show that CLaM-TTS is better than or comparable to state-of-the-art neural codec-based TTS models regarding naturalness, intelligibility, speaker similarity, and inference speed.

## 9 ACKNOWLEDGMENTS

The authors would like to thank Kangwook Lee for helpful discussions, as well as Beomsoo Kim, Gibum Seo, and Dongwon Kim for their essential support throughout the processes of data handling and evaluation of the implementation.

## 10 ETHICS STATEMENTS

CLaM-TTS is a zero-shot TTS model that leverages a pre-trained large language model, offering efficient learning and inference at a vast scale. The model's capability to produce any voice and mimic with only minimal voice input presents potential dangers, including spoofing misuse. Given the escalating risks associated with such models, it should be imperative to develop a detection model to identify audio outputs from the model and to establish a rigorous protocol for evaluating its effectiveness.

## 11 REPRODUCIBILITY STATEMENTS

For the implementation of our model, we provide Fig. 1 and description of the model architecture in Sec. 4.3 along with the hyperparameters of the model configuration in Tab. 7. To ensure the reproducibility of our experiments, we also share details, including a list and statistics of our training data in Sec. 5 and Appx. B.1, data preprocessing procedures in Appx. B.2 and Appx. B.3, training configuration and the evaluation methodology in Sec. 5. If our potential legal concerns can be addressed, we are prepared to progressively disclose, for research purposes, the inference code, pre-trained weights, and ultimately, the full training implementation.

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

## A  VARIATIONAL LOWER BOUND

We have

$$
\begin{aligned}
\log p_\theta(\boldsymbol{c}_t|\boldsymbol{x}, \boldsymbol{c}_{<t}) &= \log \int p_\theta(\boldsymbol{c}_t, \boldsymbol{z}_t|\boldsymbol{x}, \boldsymbol{c}_{<t}) \, d\boldsymbol{z}_t \\
&\overset{(a)}{=} \log \int p_\theta(\boldsymbol{z}_t|\boldsymbol{x}, \boldsymbol{c}_{<t})p_\psi(\boldsymbol{c}_t|\boldsymbol{z}_t) \, d\boldsymbol{z}_t \\
&= \log \int p_\theta(\boldsymbol{z}_t|\boldsymbol{x}, \boldsymbol{c}_{<t})\frac{p_\psi(\boldsymbol{z}_t|\boldsymbol{c}_t)p(\boldsymbol{c}_t)}{\sum_{\boldsymbol{c}_t'} p_\psi(\boldsymbol{z}_t|\boldsymbol{c}_t')p(\boldsymbol{c}_t')} \, d\boldsymbol{z}_t \\
&\overset{(b)}{=} \log \int p_\theta(\boldsymbol{z}_t|\boldsymbol{x}, \boldsymbol{c}_{<t})\frac{p_\psi(\boldsymbol{z}_t|\boldsymbol{c}_t)}{\sum_{\boldsymbol{c}_t'} p_\psi(\boldsymbol{z}_t|\boldsymbol{c}_t')} \, d\boldsymbol{z}_t \\
&\overset{(c)}{\geq} \int p_\psi(\boldsymbol{z}_t|\boldsymbol{c}_t) \left[ \log p_\theta(\boldsymbol{z}_t|\boldsymbol{x}, \boldsymbol{c}_{<t}) - \log \sum_{\boldsymbol{c}_t'} p_\psi(\boldsymbol{z}_t|\boldsymbol{c}_t') \right] d\boldsymbol{z}_t,
\end{aligned}
\tag{7}
$$

where $(a)$ follows from the modeling assumption that $\boldsymbol{c}_t$ is independent of $\boldsymbol{x}$ and $\boldsymbol{c}_{<t}$ given $\boldsymbol{z}_t$; $(b)$ follows from the assumption that $p(\boldsymbol{c}_t)$ is uniformly distributed for all $t$; and $(c)$ follows by Jensen's inequality.

We assume that an unobserved discrete random variable $k$ is involved in generating the latent vector $\boldsymbol{z}_t$ by some random process.

$$
\begin{aligned}
\log p_\theta(\boldsymbol{z}_t|\boldsymbol{x}, \boldsymbol{c}_{<t}) &= \log \sum_k p_\theta(\boldsymbol{z}_t, k|\boldsymbol{x}, \boldsymbol{c}_{<t}) \\
&= \log \sum_k p_\theta(k|\boldsymbol{x}, \boldsymbol{c}_{<t})p_\theta(\boldsymbol{z}_t|\boldsymbol{x}, \boldsymbol{c}_{<t}, k) \\
&= \log \sum_k q(k|\boldsymbol{x}, \boldsymbol{c}_{\leq t})\frac{p_\theta(k|\boldsymbol{x}, \boldsymbol{c}_{<t})p_\theta(\boldsymbol{z}_t|\boldsymbol{x}, \boldsymbol{c}_{<t}, k)}{q(k|\boldsymbol{x}, \boldsymbol{c}_{\leq t})} \\
&\geq \sum_k q(k|\boldsymbol{x}, \boldsymbol{c}_{\leq t}) \left[ \log p_\theta(\boldsymbol{z}_t|\boldsymbol{x}, \boldsymbol{c}_{<t}, k) + \log \frac{p_\theta(k|\boldsymbol{x}, \boldsymbol{c}_{<t})}{q(k|\boldsymbol{x}, \boldsymbol{c}_{\leq t})} \right].
\end{aligned}
\tag{8}
$$

By incorporating this relation into Eq. 7, we have:

$$
\log p_\theta(\boldsymbol{c}_t|\boldsymbol{x}, \boldsymbol{c}_{<t})
$$

$$
\geq \int p_\psi(\boldsymbol{z}_t|\boldsymbol{c}_t) \left[ \log p_\theta(\boldsymbol{z}_t|\boldsymbol{x}, \boldsymbol{c}_{<t}) - \log \sum_{\boldsymbol{c}_t'} p_\psi(\boldsymbol{z}_t|\boldsymbol{c}_t') \right] d\boldsymbol{z}_t
$$

$$
\geq \sum_k q(k|\boldsymbol{x}, \boldsymbol{c}_{\leq t}) \left[ \int p_\psi(\boldsymbol{z}_t|\boldsymbol{c}_t) \log \frac{p_\theta(\boldsymbol{z}_t|\boldsymbol{x}, \boldsymbol{c}_{<t}, k)}{\sum_{\boldsymbol{c}_t'} p_\psi(\boldsymbol{z}_t|\boldsymbol{c}_t')} \, d\boldsymbol{z}_t + \log \frac{p_\theta(k|\boldsymbol{x}, \boldsymbol{c}_{<t})}{q(k|\boldsymbol{x}, \boldsymbol{c}_{\leq t})} \right]
$$

$$
= \mathbb{E}_{q(k|\boldsymbol{x}, \boldsymbol{c}_{\leq t})} \left[ \mathbb{E}_{p_\psi(\boldsymbol{z}_t|\boldsymbol{c}_t)} \left[ \log \frac{p_\theta(\boldsymbol{z}_t|\boldsymbol{x}, \boldsymbol{c}_{<t}, k)}{p_\psi(\boldsymbol{z}_t|\boldsymbol{c}_t)} \frac{p_\psi(\boldsymbol{z}_t|\boldsymbol{c}_t)}{\sum_{\boldsymbol{c}_t'} p_\psi(\boldsymbol{z}_t|\boldsymbol{c}_t')} \right] + \log \frac{p_\theta(k|\boldsymbol{x}, \boldsymbol{c}_{<t})}{q(k|\boldsymbol{x}, \boldsymbol{c}_{\leq t})} \right]
$$

$$
= \mathbb{E}_{q(k|\boldsymbol{x}, \boldsymbol{c}_{\leq t})}[-D_{KL}(p_\psi(\boldsymbol{z}_t|\boldsymbol{c}_t)||p_\theta(\boldsymbol{z}_t|\boldsymbol{x}, \boldsymbol{c}_{<t}, k))] - D_{KL}(q(k|\boldsymbol{x}, \boldsymbol{c}_{\leq t})||p_\theta(k|\boldsymbol{x}, \boldsymbol{c}_{<t})) + \mathcal{B}(\psi, c_t),
$$

where $\mathcal{B}(\psi, c_t) = \mathbb{E}_{p_\psi(\boldsymbol{z}_t|\boldsymbol{c}_t)} \left[ \log \frac{p_\psi(\boldsymbol{z}_t|\boldsymbol{c}_t)}{\sum_{\boldsymbol{c}_t'} p_\psi(\boldsymbol{z}_t|\boldsymbol{c}_t')} \right] = \mathbb{E}_{p_\psi(\boldsymbol{z}_t|\boldsymbol{c}_t)}[\log p_\psi(\boldsymbol{c}_t|\boldsymbol{z}_t)] \leq 0.$

## B  ADDITIONAL DETAILS OF EXPERIMENT

### B.1  DATASET STATISTICS

The training datasets for each language are as follows:

**English**: 1) Multilingual LibriSpeech (MLS) (Pratap et al., 2020), a multi-speaker and multilingual transcribed speech dataset sourced from LibriVox audiobooks. 2) GigaSpeech (Chen et al., 2021) consists of multi-domain speeches, such as audiobooks, podcasts and YouTube along with human transcriptions. The audio dataset includes multiple speakers but lacks associated speaker information. 3) LibriTTS-R (Koizumi et al., 2023) is a restored version of the LibriTTS (Zen et al., 2019) corpus which shares the identical metadata with LibriTTS. 4) VCTK (Veaux et al., 2016) and 5) LJSpeech (Ito & Johnson, 2017) are multi-speaker and single-speaker English datasets, respectively, widely used in speech synthesis community.

**Korean**: 1) AIHub 14[8] features recordings of everyday people reading provided script sentences. 2) AIHub 15[9] has recordings of 50 professional voice actors for seven emotions (joy, surprised, sad, angry, scared, hate, neutral), five speaking styles (narrating, reading, news-like, dialogic, broadcasting), and three vocal ages (kid, young, old). 3) KsponSpeech (Bang et al., 2020) consists of 2,000 speakers and each recording has an individual freely talking about various topics in a quiet environment. The transcription follows specific guidelines regarding laughing, breathing, and a few more.

**Chinese**: WeNetSpeech (Zhang et al., 2022) is similar to GigaSpeech, comprising multi-domain speeches without speaker information.

**Japanese**: 1) ReazonSpeech (Fujimoto) is a labelled dataset made up of roughly 19,000 hours of broadcasted speech. It involves multiple speakers without speaker information. 2) KokoroSpeech (Iida, 2021) contains recordings of a single speaker reading 14 novel books.

**Others**: The datasets feature seven language subsets from MLS: German, Dutch, French, Spanish, Italian, Portuguese, and Polish.

Tab. 6 shows detailed statistics of each dataset. For the LJSpeech, VCTK, KokoroSpeech, and ReazonSpeech datasets, 99% of the entire dataset is used for training. If a specific training set is predefined, we use it as-is.

## B.2 DATA PRE-PROCESSING

Note that Gigaspeech, WeNetSpeech, and ReazonSpeech do not provide speaker labels. For the datasets, we exclude audio instances that contain two or more speakers using an open-source speaker diarization model [10]. We also compute the SNR of the audios in three datasets using waveform amplitude distribution analysis (WADA) (Kim & Stern, 2008). Audios with WADA-SNR $> 20$dB are only included in our training set.

We preprocess the large datasets to efficiently store and iterate over them. Audio metadata is first constructed in the parquet[11] format. Parquet, storing data column-wise, offers high compression rates and reduces I/O overhead, making it suitable for metadata storage. Parquet contains speaker attributes (name, gender, accent, emotion, and age), audio path, length, sample rate, and text. The constructed metadata is combined with audio data read as byte streams to form datasets using web-datasets[12]. Specifically, audio byte streams are paired with JSON data from parquet, and every 10k pairs are stored as a single TAR file. Storing data in this manner facilitates the addition of metadata items or text forms (e.g., phoneme, normalized text) in the future.

We utilize the stored speaker metadata as a part of the text prompt. The speaker attributes are appended in front of each text. One example would look like the following.

```
man, old, neutral:  We have to reduce the number of plastic bags.
```

Please check our demo page [13] for text prompting applications.

---

[8]https://www.aihub.or.kr/aihubdata/data/view.do?currMenu=115&topMenu=100&aihubDataSe=realm&dataSetSn=542

[9]https://www.aihub.or.kr/aihubdata/data/view.do?currMenu=115&topMenu=100&aihubDataSe=realm&dataSetSn=466

[10]https://huggingface.co/pyannote/speaker-diarization-2.1

[11]https://parquet.apache.org/

[12]https://webdataset.github.io/webdataset/

[13]https://clam-tts.github.io

Table 6: Statistics of datasets for different languages. We blank the number of speakers when the original dataset doesn't provide speaker information.

| Lang | Dataset | Train / Total (hrs) | # Speakers (Train) | Rate (Hz) |
|---|---|---|---|---|
| English | MLS | 44,659.74 / 44,691.05 | 5,490 | 16,000 |
| | GigaSpeech | 9,997.82 / 10,050.65 | - | 16,000 |
| | LibriTTS-R | 730.00 / 769.97 | 2,456 | 24,000 |
| | VCTK | 40.63 / 41.04 | 109 | 48,000 |
| | LJSpeech | 23.68 / 23.92 | 1 | 22,050 |
| Korean | AIHub 14 | 8,086.04 / 9,110.43 | 3,495 | 48,000 |
| | AIHub 15 | 836.78 / 951.98 | 50 | 48,000 |
| | Ksponspeech | 965.15 / 975.54 | 2,000 | 16,000 |
| Chinese | WeNetSpeech | 10,005.41 / 10,063.67 | - | 16,000 |
| Japanese | ReazonSpeech | 18,846.81 / 19,037.18 | - | 16,000 |
| | KokoroSpeech | 58.11 / 58.69 | 1 | 22,050 |
| German | MLS German | 1,966.51 / 1,995.08 | 176 | 16,000 |
| Dutch | MLS Dutch | 1,554.24 / 1,579.76 | 40 | 16,000 |
| French | MLS French | 1,076.58 / 1,096.72 | 142 | 16,000 |
| Spanish | MLS Spanish | 917.68 / 937.68 | 86 | 16,000 |
| Italian | MLS Italian | 247.38 / 257.83 | 65 | 16,000 |
| Portuguese | MLS Portuguese | 160.96 / 168.35 | 42 | 16,000 |
| Polish | MLS Polish | 103.65 / 107.87 | 11 | 16,000 |

### B.3 APPROXIMATION FOR AUDIO RESAMPLING

We pre-process the audio dataset to create mel-spectrograms with the same resolution. We first revisit the audio resampling process before describing our method. Let us denote the resampled audio, its sample rate, and its corresponding mel-spectrogram as $A_{target}$, $S_{target}$, and $M_{target}$ respectively. The original audio and its sample rate can be labelled as $A_{source}$ and $S_{source}$. Please note that our proposed model uses the mel-spectrogram as both input and output, making $M_{target}$ our final goal for the data processing. First, we apply the STFT to $A_{source}$ to obtain frequency domain components. We then perform linear interpolation in the frequency domain to adjust the number of samples per second to match $S_{target}$. Due to the occurrence of aliasing, a Low-pass filter is applied. Next, we use the inverse STFT (ISTFT) to acquire the upsampled $A_{target}$. Finally, we apply the STFT again to derive $M_{target}$.

Our approach, in contrast, is as follows: To determine $M_{target}$, we adjust the pre-set FFT size (which is also the same as window size) and hop size according to the ratio of $S_{source}$ to $S_{target}$. Using the value, we produce a linear spectrogram and then apply a mel-filter bank to generate a mel-spectrogram. We refer to it as $\tilde{M_{target}}$ and regard it as an approximation for $M_{target}$. Unlike $M_{target}$, the process of obtaining $\tilde{M_{target}}$ does not actually change the number of audio samples, which can lead to inaccuracies at high frequencies. However, due to the nature of mel-spectrograms, it can be disregarded in regions of important low-frequency features. The advantage of deriving $\tilde{M_{target}}$ is that we only need to apply the STFT once, and there's no need to store any audio in between. It allowed us to use datasets with different sample rates without time and storage consumption due to audio resampling.

In the mini-batch training scheme, we first randomly sample the amount of data up to the pre-defined maximum audio length per batch from the train set and sort them by sample rate. Then we determine the ratio of each data sample rate over the target to modify FFT size (window size) and hop size, respectively. The mel-spectrogram calculated by these STFT parameters is input to our model.

### B.4 DETAILS OF ENGLISH-ONLY EVALUATION

We average the result over three repetitions of each experiment with a randomly selected set of prompts per trial. In WER and CER evaluation on *continuation* task, we include prompt reconstruction at the beginning of the generated speech. Meanwhile, we exclude reconstructed prompts in SIM

evaluation on the same task. Recall that our model takes the speech prompt together with the corresponding transcript. In *cross-sentence* task, we use Montreal Forced Aligner (MFA) (McAuliffe et al., 2017) to align transcript and audio. We randomly select a starting point at the beginning of a word in audio and use subsequent 3-second audio for baselines. Otherwise, we use the audio and text that include as many words as possible within the subsequent 3-second duration. This yields audio clips whose average length is around 2.7 seconds. We name it as *word-based* prompting.

## B.5 SUBJECTIVE EVALUATION

We carry out evaluations using Amazon Mechanical Turk (MTurk)[14]. In QMOS, we direct evaluators to assess the quality and clarity of each recording, considering sound quality and clarity of speech. For SMOS, evaluators gauge the likeness of samples to the provided speech prompts, taking into account the speaker similarity, style, acoustics, and background disturbances. In CMOS, evaluators compare overall quality of a synthesized sample to that of a reference. Using the given scale, they judge whether the synthesized version was superior or inferior to the reference.

QMOS and SMOS employ a 1 to 5 scale of integer, where 5 signifies top quality. CMOS uses a scale from -3 (indicating the synthesized speech is much worse than the reference) to 3 (indicating it's much better), with 1-unit intervals. For QMOS, SMOS, and CMOS, samples garner 10, 6, and 12 ratings respectively. We omit evaluators whose average scores deviate by two standard deviations or more from the mean over every evaluator. All evaluators are US-based.

Table 7: The detailed model configurations of Mel-VAE.

| Module | Configuration | Value |
|---|---|---|
| Encoder | hidden size | 256 |
| | channel multiplication | [1, 1, 2, 2] |
| | dropout | 0.0 |
| Decoder | hidden size | 256 |
| | channel multiplication | [1, 1, 2, 2] |
| | dropout | 0.0 |
| | ConvNeXt hidden layer | 80 |
| Probabilistic RVQ | Depth | 32 |
| | Vocab Size | 1024 |
| | Channel Size | 512 |

Table 8: The detailed model configurations of Text-to-Code.

| Module | Configuration | Value |
|---|---|---|
| Encoder | hidden size | 1536 |
| | Number of Heads | 16 |
| | Number of Layers | 36 |
| | Feedforward Dimension | 3840 |
| | dropout | 0.1 |
| Decoder | hidden size | 1536 |
| | Number of Heads | 16 |
| | Number of Layers | 12 |
| | Feedforward Dimension | 3840 |
| | dropout | 0.1 |
| Latent MoG | Weight Predictor Dimension (k) | 2048 |
| | label smoothing | 0.01 |

---

[14] https://www.mturk.com/

## C MODEL CONFIGURATIONS AND IMPLEMENTATION DETAILS

### C.1 MODEL CONFIGURATIONS

We provide detailed model configurations in Tab. 7 and 8.

### C.2 IMPLEMENTATION DETAILS

**MelVAE** Our embedding vector of residual vector quantization is structured to decrease in length with each increasing depth $d$, denoted as $e_\psi(\boldsymbol{c}_{t,d}; d)$. This configuration ensures that embeddings at higher depths capture more detailed features. The formulation for this parameterization is given by:

$$
\begin{aligned}
&e_\psi(\boldsymbol{c}_{t,d}; d) \\
&= \alpha_d \frac{\tilde{e}_\psi(\boldsymbol{c}_{t,d}; d)}{\|\tilde{e}_\psi(\boldsymbol{c}_{t,d}; d)\|}, \\
&\text{where} \quad \alpha_d = \exp(max\_scale\_logit_\psi) \sum_{i=d}^{D} \text{softmax}(scale\_logits_\psi)_i.
\end{aligned}
\tag{9}
$$

Here, $\alpha_d$ dynamically adjusts the scale of embedding vectors at the depth $d$, with the trainable parameters of the residual vector quantizer being $max\_scale\_logit_\psi$, $scale\_logits_\psi$, and $\tilde{e}_\psi(\boldsymbol{c}_{t,d}; d)$.

**Latent MoG** We present an efficient approach to managing the number of mixtures, $k$, in the Mixture of Gaussians (MoG) of our latent language model. The dimensionality of the MoG output is directly proportional to both $k$ and the dimension $m$ of the mean vector $\mu_\theta(k, x, \boldsymbol{C}_{<t}) \in \mathbb{R}^m$. This dimensionality can become unwieldy as $k$ increases. To address the challenge of managing the excessive size of the output weight, we employ a strategy to compress it. This approach yields a low-rank prediction $\tilde{\mu}_\theta(k, x, \boldsymbol{C}_{<t}) \in \mathbb{R}^n$ and a projection matrix $M \in \mathbb{R}^{m \times n}$, where $n < m$, effectively enabling us to represent $\mu_\theta(k, x, \boldsymbol{C}_{<t})$ as $M\tilde{\mu}_\theta(k, x, \boldsymbol{C}_{<t})$. This compression facilitates more efficient computation of the training loss, which involves calculating the L2 distance between the mean vector $\mu_\theta(k, x, \boldsymbol{C}_{<t})$ and the quantized latent representation of speech $\hat{\boldsymbol{z}}_t$ as $\|\hat{\boldsymbol{z}}_t - \mu_\theta(k, x, \boldsymbol{C}_{<t})\|^2$. The equation is expanded as follows:

$$
\begin{aligned}
&\|\hat{\boldsymbol{z}}_t - \mu_\theta(k, x, \boldsymbol{c}_{<t})\|^2 \\
&= \|\hat{\boldsymbol{z}}_t - M\tilde{\mu}_\theta(k, x, \boldsymbol{c}_{<t})\|^2 \\
&= \hat{\boldsymbol{z}}_t^T \hat{\boldsymbol{z}}_t + \tilde{\mu}_\theta(k, x, \boldsymbol{c}_{<t})^T (M^T M) \tilde{\mu}_\theta(k, x, \boldsymbol{c}_{<t}) - 2(M^T \hat{\boldsymbol{z}}_t)^T \tilde{\mu}_\theta(k, x, \boldsymbol{c}_{<t}).
\end{aligned}
\tag{10}
$$

Precomputing $M^T M$ and $M^T \hat{\boldsymbol{z}}_t$ in Eq. 10 significantly reduces memory usage during L2 distance computation. For numerical stability, we apply spectral normalization (Miyato et al., 2018) to $M$. In practice, we use all these techniques when $k > 512$.

## D ADDITIONAL RESULTS

### D.1 DURATION-BASED PROMPTING

In *duration-based* prompting, we randomly pick one utterance from the target speaker and choose a 3-second segment. We then use the same model from Sec. 5, which is OpenAI's Whisper, to transcribe this segment, and input this audio-text pair as a prompt. However, this method introduces transcription errors from Whisper, causing our model to include incorrect text at the beginning of generated speech and hence to have poor WER and CER performances.

Tab. 9 shows the results that *word-based* prompting is more effective than *duration-based* in our character-based Text-to-Code model.

### D.2 COMPARISON OF T5 VARIANTS

We compare the performance of the available T5 variants: 1) T5, which approaches every NLP task as a text-to-text conversion, regardless of whether it's translation, summarization, or question-answering. 2) mT5 (Xue et al., 2021b), a multilingual variant of T5 handling multiple languages

Table 9: Results of comparison between *(word-based) cross-sentence* and *(duration-based) cross-sentence* task.

| Model | WER | CER | SIM-o |
|---|---|---|---|
| CLaM-en (*word-based*) from Tab. 2 | 5.11 | 2.87 | 0.4951 |
| CLaM-en (*duration-based*) | 8.68 | 5.69 | 0.5026 |

Table 10: Results of Multilingual *continuation* task. The scores of WeNetSpeech are absent because our ASR tool doesn't recognize the generated Chinese speech. Since Japanese lacks spacing, WER measurement is not feasible.

| Lang | Dataset | WER↓ | CER↓ | SIM-o↑ |
|---|---|---|---|---|
| English | GigaSpeech | 9.75 | 2.86 | 0.3738 |
| | LibriTTS-R | 3.12 | 0.96 | 0.5112 |
| | VCTK | 1.48 | 0.73 | 0.3849 |
| | LJSpeech | 6.55 | 4.75 | 0.5879 |
| English (HuBERT) | GigaSpeech | 16.90 | 4.74 | 0.3738 |
| | LibriTTS-R | 4.33 | 0.74 | 0.5112 |
| | VCTK | 5.26 | 1.98 | 0.3849 |
| | LJSpeech | 7.65 | 3.10 | 0.5879 |
| Korean | AIHub 14 | 20.21 | 1.80 | 0.5423 |
| | AIHub 15 | 13.08 | 2.35 | 0.5280 |
| | Ksponspeech | 30.24 | 20.02 | 0.4488 |
| Chinese | WeNetSpeech | - | - | 0.2600 |
| Japanese | ReazonSpeech | - | 49.37 | 0.2699 |
| | KokoroSpeech | - | 11.46 | 0.5653 |

within one model. 3) ByT5, which is trained on byte sequences rather than subword tokens. 4) Flan-T5 (Chung et al., 2022), which employs prompting for pre-training. 5) T5-lm-adapt, another T5 model pre-trained on denoising and fine-tuned as a prefix language model. And to compare the case where the Text-to-Code model receives phoneme sequence as an input, we employ 6) a phoneme encoder XPhoneBert (The Nguyen et al., 2023) that is of the same size with an encoder of t5-base. We train only the decoders with identical decoder structures across both settings. For each variant, we implement Text-to-Code with each *base* architecture and train it on the same Mel-VAE code following Sec. 4.2. All models are initialized using pre-trained weights. Then, we evaluate the models in a continuation task, measuring WER, CER, and SIM-o. We use the MLS English subset for training and follow the training procedures described in Sec. 5. Evaluations are conducted on the LibriSpeech test-clean set as outlined in Sec. 6.1.

Tab. 11 shows that ByT5 outperforms the other variants of the T5 model except that the phoneme model. Notably, ByT5 significantly surpasses T5 by merely changing the input format. Flan-T5 and T5-lm-adapt also demonstrate better performance than T5 by pre-training and fine-tuning as a prefix language model. The results imply that fine-tuning ByT5 as a prefix language model might yield even better results if then trained as Text-to-Code. The superior results of phoneme variant demonstrate why many TTS studies have preferred phonemes over characters. Despite its remarkable performance, we choose byT5 as the base model. The reason for this is that while large-scale pretrained models based on text, like byT5, have been released and researched, phoneme-based encoders have seen only small-scale models made available and studied due to the limited phoneme data. It is still understood that using phonemes as input is effective, and both text and phonemes can be selectively chosen based on their scalability and robustness. We leave this exploration for future work.

## D.3    EFFECTS OF CODEWORD EMITTING RATE

We conducted additional ablation studies to analyze the impact of varying codeword rates in the proposed framework. The results demonstrate a trade-off: reducing the code emitting frequency

Table 11: Results of *continuation* task of T5 variants. All models are *base* model.

| Model | WER↓ | CER↓ | SIM-o↑ |
|-------|------|------|--------|
| ByT5 | 2.79 | 1.00 | 0.3879 |
| T5-lm-adapt | 2.88 | 1.17 | 0.3821 |
| Flan-T5 | 2.92 | 1.21 | 0.3802 |
| mT5 | 4.62 | 2.56 | 0.3762 |
| T5 | 9.48 | 7.04 | 0.3634 |
| T5-phoneme | **2.62** | **0.96** | **0.3943** |

degrades audio quality while increasing it diminishes language modeling performance. This finding shows that we chose the codeword rate that operates at a sweet spot in this trade-off.

**The quality of reconstructed audios from Mel-VAE**: We compared the performances of Mel-VAEs trained with different codeword rates. The codeword rate in each Mel-VAE was adjusted according to the downsampling factor when generating latent representations from mel-spectrograms. We employed downsampling factors: 16, 8 (ours), and 4; and measured PESQ and ViSQOL of the reconstructed audio from each model. Tab. 12 shows that the audio reconstruction quality of the Mel-VAEs deteriorates as the downsampling factor increases.

Table 12: Results of audio reconstruction of different Mel-VAEs.

| Model | PESQ↑ | ViSQOL↑ |
|-------|-------|---------|
| Mel-VAE-df8 (default) | 2.95 | 4.66 |
| Mel-VAE-df4 | 3.10 | 4.74 |
| Mel-VAE-df16 | 2.42 | 4.35 |

**Text-to-code language model performances**: We trained identical latent language models to generate codes for each of the Mel-VAEs and measured WER, CER, and speaker similarity. Similar to the trend in reconstructed audio quality, Tab. 13 shows that a higher code emitting frequency tends to increase speaker similarity. However, intelligibility, measured by WER and CER, performed better with a 16-fold compression compared to a 4-fold compression. This suggests that the longer the code length predicted, the more challenging it is for latent language models. Moreover, the default setting of an 8-fold compression shows the best intelligibility, indicating that our setting finds a sweet spot in balancing audio quality and latent language model prediction in the trade-off. We also reported the end-to-end inference time of 10s speech, varying with frequency.

Table 13: Results of *continuation* task and the inference time of different Mel-VAEs.

| Model | WER↓ | CER↓ | SIM-o↑ | Inference Time↓ |
|-------|------|------|--------|-----------------|
| ByT5-base-df8 (default) | 2.79 | 1.00 | 0.3879 | 3.46s |
| ByT5-base-df4 | 4.56 | 2.32 | 0.4117 | 6.87s |
| ByT5-base-df16 | 3.20 | 1.19 | 0.3629 | 1.88s |

## D.4 Effects of data scale

The results in Tab. 14 show that the performance of our model improves with an increase in the volume of training data. Specifically, we trained ByT5-base models on different sizes of training datasets: 1. (small dataset) 50% of the MLS English subset; 2. (normal dataset) the full MLS English subset; and 3. (large dataset) a comprehensive English dataset that includes MLS English subset, Gigaspeech, LibriTTS-R, VCTK, and LJSpeech. We observed a noticeable degradation in performances (WER, CER, and speaker similarity) when the small dataset was employed. Conversely, training on the large dataset resulted in a performance enhancement in speaker similarity while maintaining WER and CER compared to the model trained on the normal dataset. This evidence strongly indicates that scaling up training data can significantly boost our model's performance.

Table 14: Results of *continuation* task of different dataset sizes.

| Dataset | WER↓ | CER↓ | SIM-o↑ |
|---|---|---|---|
| Small (22K hr) | 3.06 | 1.15 | 0.3851 |
| Normal (44K hr) | 2.79 | 1.00 | 0.3879 |
| Large (55K hr) | 2.79 | 0.98 | 0.4001 |

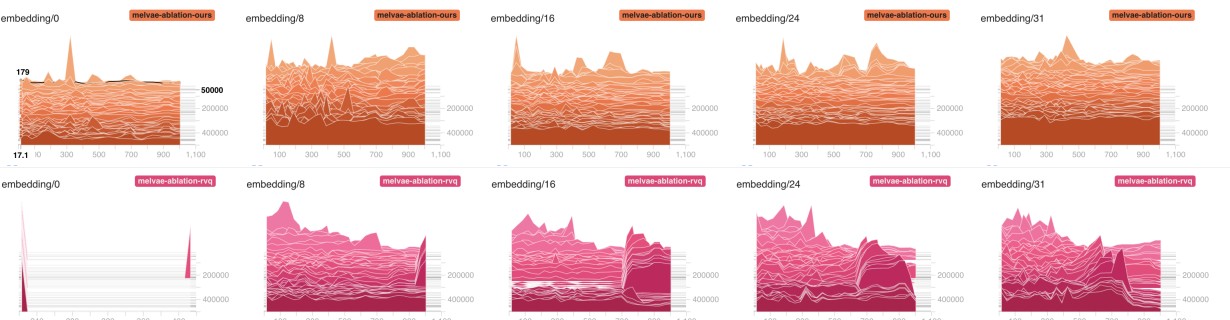

Figure 2: The codebook usages of the probabilistic RVQ and prior RVQ method during training. The code book usages are plotted for the 0th, 8th, 16th, 24th, and 31st depths, from left to right.

