# OpenReview forum: "CLaM-TTS: Improving Neural Codec Language Model for Zero-Shot Text-to-Speech"
_ICLR.cc/2024/Conference — ICLR 2024 poster_

### Official Review · Reviewer_D6DJ · 2023-11-01

**Soundness:** 3 good
**Presentation:** 4 excellent
**Contribution:** 4 excellent
**Rating:** 8
**Confidence:** 4

**Summary:**

This paper focuses on the LLM-based zero-shot TTS task. The authors introduce CLaM-TTS, which uses probabilistic residual vector quantization for two main benefits: it significantly compresses token length and enables the generation of multiple tokens simultaneously, eliminating the need for cascaded modeling. Experimental results show that CLaM-TTS performs on par with or better than existing zero-shot TTS baselines in terms of naturalness, intelligibility, speaker similarity, and inference speed. The study also explores how the pretraining extent of language models and their text tokenization strategies affect performance.

**Strengths:**

- The paper is commendably clear and straightforward, making it accessible to its readers. All the training details are meticulously presented, contributing to the paper's clarity and ease of comprehension.

- While the paper adheres to the conventional structure of token + language modeling (LM), the introduction of the CLaM approach marks a significant innovation. CLaM addresses two critical issues in the field, demonstrating sufficient novelty and contributing meaningfully to the existing body of research.

- The experiments conducted are comprehensive, effectively demonstrating the efficacy of the proposed methods. The completeness of these experiments lends credibility to the findings and supports the conclusions drawn in the paper.

- In terms of RTF, the CLaM-TTS model exhibits superior speed compared to the Voicebox and Vall-E TTS models. This enhanced performance highlights the advantages of the proposed approach.

**Weaknesses:**

It's commendable that the paper is well-organized and well-written, making it accessible and comprehensible to its audience. I don't have a comment on the weakness.

**Questions:**

- It's mentioned by the authors that different codeword rates can be compared. Including a detailed analysis of the tradeoff between RTF and performance in relation to varying codeword rates would greatly enhance the paper. Such an analysis would provide a clearer understanding of how changes in codeword rate impact both the efficiency and effectiveness of the model, offering valuable insights for optimizing model performance in practical applications.

---

> ### Author Response · Authors · 2023-11-21
> **Author’s Response to Reviewer D6DJ**
>
> We sincerely appreciate your positive review and constructive feedback.
>
> > `Regarding the tradeoff between efficiency and effectiveness in varying codeword rate`
>
> As per your great suggestion, we conducted an ablation study to analyze the impact of varying codeword rates in the proposed system and addressed it in the first section of “General Response to AC and Reviewer”. The experimental results demonstrate a trade-off: *reducing the code emitting frequency degrades audio quality while increasing it diminishes language modeling performance*. We also reported on the inference speed of the entire system in relation to varying code frequencies. This finding illustrates that our main model operates at a balance point within this trade-off, efficiently managing both aspects.

---

> > ### Comment · Reviewer_D6DJ · 2023-11-23
> >
> > I don't have further questions. Thank you for the response.

---

### Official Review · Reviewer_CLFG · 2023-11-01

**Soundness:** 4 excellent
**Presentation:** 3 good
**Contribution:** 4 excellent
**Rating:** 8
**Confidence:** 4

**Summary:**

This work presents a language model based approach to generate multiple neural codec tokens for zero-shot text-to-speech. This work proposed a probabilistic residual vector quantizer, which allows the language model to generate multiple neural codes at the same time, which considerably reduces the sequence length in modeling and speed up the inference time. The results demonstrate the effectiveness of the proposed method, in terms of naturalness, intelligibility and speaker similarity.

**Strengths:**

This paper proposed a probabilistic RVQ quantizer, which enables the language model to emit multiple neural codes at each inference step. This address the long inference sequence issue in the previous methods like Vall-E and audioLm. The authors have provided a detailed derivation of how the RVQ and latent language modeling, which is very helpful to understand this method.

**Weaknesses:**

It is not very clear to me why the probabilistic RVQ can lead to much compressed token sequence. For example, audiolm uses the 50Hz semantic token, while this work use a 10Hz codeword rate. It would be great if the author can explain or provide experimental results showing the effect of codeword emitting frequency.

**Questions:**

What's y and \hat{y} in Eq(7)? It looks like it is never defined?

---

> ### Author Response · Authors · 2023-11-21
> **Author’s Response to Reviewer CLFG**
>
> We would like to thank the reviewer for providing a positive review and spotting unclear points.
>
> > `Regarding the effect of codeword emitting frequency`
>
> Thank you for pointing out the parts in our paper that were not explained clearly. The semantic tokens in AudioLM are a sequence of 50Hz, while our tokens consist of 32 sequences of 10Hz due to the use of 32-depth residual vector quantization. This means that when representing audio, we can achieve a higher "length" compression rate compared to AudioLM by compressing the audio into multiple sequences using RVQ. We have conducted experiments and evaluations regarding the codeword emitting frequency, which we addressed in the first section of “General Response to AC and Reviewer”. These results demonstrate that our main model operates in the sweet spot for the trade-off. We sincerely request you to review them for additional context.
>
> > `Definitions of $y$ and $\hat{y}$`
>
> For the definitions of the variables, please refer to the second paragraph in Sec. 3.2.1. $y$ and $\hat{y}$ denote a mel-spectrogram and its reconstruction from the decoder, respectively.

---

### Official Review · Reviewer_ZbF5 · 2023-11-07

**Soundness:** 2 fair
**Presentation:** 2 fair
**Contribution:** 2 fair
**Rating:** 5
**Confidence:** 5

**Summary:**

In this paper, the authors propose a TTS model(CLaM-TTS) with a probabilistic residual vector quantization. They achieve superior compression in token length and allow model to generate multiple tokens at once. They evaluate the proposed CLaM-TTS in continuation and cross sentence tasks on English and Multilingual respectively. The experimental results show the model can bring clear improvements compared to the base RVQ and show its ability on both tasks.

**Strengths:**

1.The paper is clear and well written. The article also offers a comprehensive mathematical analysis of using detailed mathematical derivation to illustrate the probabilistic residual vector quantization of the article.

2.The paper completed a good ablation experiment, not only verified the speech quality produced with the RVQ used in the model, but also make comparison of different T5 model to illustrate the importance of pretrained language model.

**Weaknesses:**

1. Novelty. The VAE or latent diffusion models are not something new, and thus the overall novelty could be questioned.

2. Experimental comparison. The authors mostly compare their work with the outdated model YourTTS in English Librispeech, or without comparison in multilingual TTS. Thus, some competing models are better to include. The demos are too limited, it is recommended to include demo audios from other work for comparison.

3. Results. In most tasks, the model cannot achieve state-of-the-art results but claims that in the paper, and thus the performance can be questioned.

**Questions:**

/

---

> ### Author Response · Authors · 2023-11-21
> **Author’s Response to Reviewer ZbF5**
>
> We would like to thank the reviewer for providing constructive feedback and spotting unclear points.
>
> > `Contribution of this paper`
>
> To address the reviewer's concerns and clarify our methodology, we emphasize the distinctions between our approach and similar existing generative methodologies, VQGAN and RQ-Transformer, among generative models such as VAE and latent diffusion.
>
> 1. **Introduction and utilization of probabilistic residual vector quantization**: This feature not only enhances codebook usage to improve reconstruction quality but also offers a principled approach to generate residual codes from latent language modeling within a variational framework.
> 2. **Employing a single autoregressive transformer**:  Our method proposes a process where a single autoregressive transformer can generate $N$ tokens at each decoding step by utilizing the probabilistic modeling of the residual vector quantizer, without the need for depth-transformers or non-autoregressive (NAR) transformer decoders. Our method shares similarities with VQGAN in that both involve 1) autoencoding of input data using VQVAE and 2) generation of the quantized latent representation using an autoregressive transformer. However, the vector quantizer in VQGAN employs plain vector quantization, producing a sequence of tokens, thereby enabling the transformer to model the discrete sequence with a softmax output. In contrast, RQ-Transformer and neural audio codec language models, which use $N$-depth residual vector quantization, produce $N$ sequences of tokens. This results in the need for not only an autoregressive transformer to decode along the time axis but also an additional depth-transformer to model $N$ tokens along the depth axis or a NAR transformer decoder to generate a token sequence at each decoding step.
>
>
> > `Regarding the performance of the proposed method`
>
> Thank you for your constructive criticism, which highlights the weaknesses in our research. We are truly grateful for your thorough review.
>
> We acknowledge the observed performance gap of our method compared to Voicebox, particularly in cross-sentence tasks where models are required to generate outputs from discontinuous texts and partially truncated audio prompts. To address this issue, we have empirically analyzed the performance improvements when incorporating phonemes as inputs (as done in VALL-E, Voicebox, and SPEAR-TTS) and durations (as in Voicebox and YourTTS) into our model. We also discuss methods that directly train on cross-sentence tasks, such as those employed by SPEAR-TTS. These investigations are critical in understanding the reasons behind our model's performance drop in cross-sentence tasks compared to other models. In response to the concerns you raised, we have provided detailed explanations in the second section of our “General Response to AC and Reviewer”.
>
> Nonetheless, it is important to emphasize why our approach uniquely relies on plain text input, unlike other approaches. This is due to our model's ability to seamlessly integrate a pre-trained large language model, which is trained on a diverse array of texts and tasks. This integration allows us to achieve performance comparable to other baselines that employ more complex modeling.
>
> Moreover, as per your great suggestion, we have included samples from other competitive models (VALL-E, NaturalSpeech 2, and Mega-TTS) on our demo page for comparison. These samples were taken from each model's demo pages. We note that the small number of these additional samples is due to the lack of official codes or pre-trained models. We acknowledge that the performance analysis using this limited number of additional samples may not have statistical significance. Nevertheless, we kindly request the reviewer to consider these additional samples in their evaluation.
>
> demo page: https://clam-tts-mos.s3.us-east-2.amazonaws.com/demo/index.html

---

### Official Review · Reviewer_yc4a · 2023-11-10

**Soundness:** 3 good
**Presentation:** 1 poor
**Contribution:** 2 fair
**Rating:** 3
**Confidence:** 4

**Summary:**

This paper proposes Clam-TTS, a new method that learns discrete latent codecs at a much lower frequency of 10hz compared to common codecs at 50-100hz. To do so, authors propose training a Mel-VAE model that predicts several "RQ" codes at once. Once this part of the network is trained, the authors propose to train latent language model with a gaussian mixture based decoder that attempt to sample codes that would match the ground truth. The codes are then passed through a decoder which outputs mel spectrograms. The paper shows that this approach can yield good performance comparable to SOTA methods on continuations and cross-sentence speech generation.

**Strengths:**

- Interesting approach to generating low frequency residual vector codes suitable to modeling in language models
- good performance on a variety of tasks
- appears to be grounded in theory

**Weaknesses:**

- voicebox significantly outperforms this work. while this not necessarily detracts from the contributions of this work, the fact that it includes duration predictions and works on phones is not a great excuse as many other tts systems use this recipe
- the paper is quite dense and difficult to follow. claims such as parallel generation of multiple tokens are seen in abstract and conclusion but finding how this is actually done is not straight forward and the description of "parallel predictors" is obtuse.
- Missing references: one of the first speech discretization and speech vector quantization techniques was proposed in Vq-wav2vec (https://arxiv.org/abs/1910.05453)
- section 5.2: ablation studty -> ablation study

overall my suggestion is to clean up the presentation to make this paper more accessible and then I believe it would be a useful contribution to the community

**Questions:**

- One of the central claims of this paper is that it manages to describe speech with discrete tokens at much lower frequency than many other approaches. While there is discussion about how lower frequency may lead to worse performance, it would be nice to see empirical ablations study of how frequency affects accuracy and performance of the system
- why are mel spectrograms necessary to obtain the desired compression? can you not progressively downsample raw wave forms to acheive the same effect?

---

> ### Author Response · Authors · 2023-11-21
> **Author’s Response to Reviewer yc4a**
>
> We would like to express our sincere gratitude for the constructive feedback and detailed suggestions, which helped us improve the manuscript. We provide point-by-point replies below.
>
> > `Regarding the performance of the proposed method compared to VoiceBox`
>
> Thank you for your constructive criticism, which highlights the weaknesses in our research. We are truly grateful for your thorough review. We acknowledge the observed performance gap of our method compared to Voicebox.
>
> In response to the concerns, we would like to clarify that our model can easily incorporate phoneme or duration to improve the performance. We have empirically analyzed the performance improvements when incorporating phonemes as inputs (as done in VALL-E, Voicebox, and SPEAR-TTS) or durations (as in Voicebox and YourTTS) into our model. We also discuss methods that directly train on cross-sentence tasks, such as those employed by SPEAR-TTS. These investigations are critical in understanding the reasons behind our model's performance drop in cross-sentence tasks compared to other models. We provide the empirical results and detailed discussion in the second section of “General Response to AC and Reviewer”.
>
> Nonetheless, it is important to emphasize why our approach uniquely relies on plain text input, unlike other approaches. This is due to our model's ability to seamlessly integrate a pre-trained large language model, which is trained on a diverse array of texts and tasks. This integration allows us to achieve performance comparable to other baselines that employ more complex modeling.
>
>
> > `Clarification of the descriptions`
>
> Thank you for pointing out the parts of our paper where the explanation was insufficient. We are grateful for your detailed feedback. In our work, the latent language model produces three distinct outputs, representing 1) the mixture weights, 2) the means of the mixture of Gaussian distribution, and 3) the probability of ending the generation. Initially, we described these three components collectively as “parallel predictors”. However, in light of your feedback, we have replaced this term in the text with the aforementioned description. Furthermore, we elucidate the decoding step in which multiple tokens are generated from the output mixture of Gaussians (MoG) distribution, consisting of the mixture weights and the means. During each decoding step, a latent vector sampled from this distribution is transformed into latent codes through the residual vector quantizer of Mel-VAE. The autoregressive decoding step concludes when the probability of ending the generation reaches or exceeds 0.5.
>
>
> > `Missing references & typos`
>
> In response to your feedback, we have added the missing reference and corrected typos, as well as other areas requiring revision, in the updated version of the paper.
>
> > `Ablation study regarding the frequency`
>
> To address the critical concern you raised, we conducted an ablation study to analyze the impact of varying codeword rates in the proposed system and addressed it in the “General Response to AC and Reviewer”. The experimental results demonstrate a trade-off: reducing the code emitting frequency degrades audio quality while increasing it diminishes language modeling performance. This finding illustrates that our main model operates at a balance point in this trade-off. We sincerely request you to review them for additional context.
>
> > `Using mel-spectrogram for the desired compression`
>
>  Our choice to use mel-spectrograms instead of raw waveforms as the input for the VAE is driven by a practical consideration. It requires substantially more memory and trainable parameters to train VAE with raw waveforms due to the extended input length. In contrast, using mel-spectrograms allows us to train VAE with considerably fewer resources as well as to use an off-the-shelf vocoder to convert mel-spectrograms to raw waveforms. Nonetheless, as the reviewer pointed out, pursuing higher reconstructed audio quality with the desired compression in an end-to-end manner with raw waveforms can be a promising direction for future work.

---

### Official Review · Reviewer_w2yt · 2023-11-10

**Soundness:** 4 excellent
**Presentation:** 4 excellent
**Contribution:** 4 excellent
**Rating:** 8
**Confidence:** 4

**Summary:**

This paper introduces CLaM-TTS, a novel text-to-speech synthesis model. CLaM-TTS combines several ideas from the literature into a single model trained on several large-scale datasets and demonstrates high-quality speech generation. The input to this model is text and the output is mel spectrograms, which must then be converted to audio waveforms using a vocoder.

The model consists of the following components:

1. Audio quantizer: They introduce Mel-VAE, a model for encoding mel spectrograms into a sequence of multi-dimensional discrete codes. The Mel-VAE decoder is used to convert the discrete codes back into mel spectrograms during inference.
2. Autoregressive synthesis network: A conditional language model is used to convert text into discrete codes. The language model is conditioned upon text (encoded by a pretrained network) and autoregressively predicts discrete audio codes.

The audio quantizer is based on residual vector quantization (RVQ). Unlike traditional language models which use a softmax output layer to predict discrete codes, the CLaM-TTS synthesis network instead uses a Gaussian mixture model (GMM) in order to predict a continuous output, which is then converted into a discretized output using the Mel-VAE RVQ quantizer.

This system is then trained on 100k hours in multiple languages and evaluated on a number of metrics, including speaker similarity, MOS, and ASR-based intelligibility and WER, demonstrating competitive performance.

**Strengths:**

This paper has clear and detailed presentation. The models are explained clearly, referencing prior literature appropriately while providing sufficient detail for readers unfamiliar with parts of it to understand the paper. Hyperparameters are stated in the appendix and reasoning for the architectural choices is included.

The key contribution of this paper, in my opinion, consists of using a Gaussian mixture model as the output distribution of the language model, using the RVQ quantizer to convert this output to a set of discrete codes at inference time. This approach elegantly avoids having to build a conditional multi-level sampling scheme, which is expensive at inference time.

The baselines used for this paper are recent and strong, and the metrics used to evaluate the result are also thorough and varied. Additionally, the authors highlight both successes and failures of their method, including baselines which outperform their model on certain metrics.

**Weaknesses:**

One of the core claims of the paper has to do with two-stage pipelines for coarse-to-fine-grained audio token generation: "A shared characteristic among these neural codec language models is their two-stage pipeline; they autoregressively generate coarse-grained audio tokens and decode them into fine- grained representations .... we design a language model that generates from coarse to fine-grained tokens without needing a two-stage pipeline."

The paper makes the claim that the continuous modeling approach they apply avoids this and yields improved quality. However, there is no side-by-side comparison which shows that. To make this paper stronger, there needs to be a comparison of CLaM-TTS with and without the two stage system. This comparison would demonstrate their point effectively.

**Questions:**

Does scaling the model to 100k hours improve model performance? It is unclear from the paper if this is actually helpful. Similarly, does scaling up the model improve performance? Although scaling characteristics are not the main purpose of the paper, it would be clearer if there were some comparisons with smaller and larger models, to demonstrate that the model is able to take advantage of the scale of the training data.

---

> ### Author Response · Authors · 2023-11-21
> **Author’s Response to Reviewer w2yt**
>
> We do appreciate your positive review and valuable feedback, and we hope our response fully addresses your concern.
>
> > `CLaM-TTS with and without the two-stage system`
>
> Thank you for your valuable comment. While our method seamlessly generates from coarse to fine-grained tokens without the necessity for a two-stage pipeline, we do not believe that modeling our approach as a two-stage system would inherently degrade its performances. We want to clarify that, in this work, we provide a method that efficiently addresses the complex modeling challenges inherent in text-to-speech synthesis systems at scale. Our proposed probabilistic residual vector quantization not only enhances the reconstruction of audio quality but also offers a principled way to generate multiple codes within a variational inference framework. This approach effectively allows for large-scale text-to-speech synthesis without adding modeling complexities or compromising inference speed.
>
> > `scaling up dataset and model sizes`
>
> Thank you for highlighting the importance of conducting additional experiments to substantiate our claims. We have carried out an ablation study. Following your recommendation, we will incorporate these results into the revised manuscript.
>
> **Scaling up dataset size**: The results in the below table show that the performance of our model improves with an increase in the volume of training data. Specifically, we trained ByT5-base models on different sizes of training datasets: 1. (*small dataset*) 50% of the MLS English subset; 2. (*normal dataset*) the full MLS English subset; and 3. (*large dataset*) a comprehensive English dataset that includes MLS English subset, Gigaspeech, LibriTTS-R, VCTK, and LJSpeech. We observed a noticeable degradation in performances (WER, CER, and speaker similarity) when the small dataset was employed. Conversely, training on the large dataset resulted in a performance enhancement in speaker similarity while maintaining WER and CER compared to the model trained on the normal dataset. This evidence strongly indicates that scaling up training data can significantly boost our model's performance.
>
> | Dataset | WER | CER | SIM-o |
> | -------- | -------- | -------- | -------- |
> | small (22K hr)  | 3.06     | 1.15| 0.3851|
> | normal (44K hr) | 2.79     | 1.00| 0.3879|
> | large (55K hr) | 2.79     | 0.98| 0.4001|
>
> **Scaling up model size**: Due to time constraints, we were unable to train ByT5-base model on the large dataset with as many steps as the trained ByT5-large model. Consequently, a precise comparison is challenging at this stage, but we share the current results. We will include the experimental results in the revised manuscript as soon as they become available.
> | Model size | WER | CER | SIM-o |
> | -------- | -------- | -------- | -------- |
> | ByT5-base      | 2.79     | 0.98| 0.4001|
> | ByT5-large     | 2.36     | 0.79| 0.4767|

---

### Author Response · Authors · 2023-11-21
**General Response to AC and Reviewer**

(R1 = R-w2yt, R2 = R-yc4a, R3 = R-ZbF5, R4 = R-CLFG, R5 = R-D6DJ)

We sincerely thank the reviewers for their thoughtful and constructive feedback. We appreciate that all reviewers acknowledge the novelty of the proposed probabilistic residual vector quantization devised in a variational framework which enables direct residual code modeling without the need for cascaded modeling (R1, R2, R4, R5) and that our paper is clear and straightforward (R1, R3, R5), and conducts comprehensive experiments to demonstrate the efficacy of the proposed method (R3, R5).

As for the concerns/questions raised, we believe that we successfully addressed all of them sufficiently and replied in line with each review. We respond to some high-level comments here in the general response.

**Criticism 1: Effects of codeword emitting rate on the system**

We do appreciate the constructive suggestion of adding the experimental results to verify the effects of varying codeword rates on the proposed approach to further clarify our claim (R2, R4, R5).

By following the suggestion, we conducted additional ablation studies to analyze the impact of varying codeword rates in the proposed framework. The results demonstrate a trade-off: *reducing the code emitting frequency degrades audio quality while increasing it diminishes language modeling performance.* This finding shows that we chose the codeword rate that operates at a sweet spot in this trade-off.

**The quality of reconstructed audios from Mel-VAE**: First, we compared the performances of Mel-VAEs trained with different codeword rates. The codeword rate in each MelVAE was adjusted according to the downsampling factor when generating latent representations from mel-spectrograms. We employed downsampling factors: 16, 8 (ours), and 4;  and measured  PESQ and ViSQOL of the reconstructed audio from each model. The below table shows that the audio reconstruction quality of the Mel-VAEs deteriorates as the downsampling factor increases.


| Model | PESQ | ViSQOL |
| -------- | -------- | -------- |
| Mel-VAE-df8 (default)     | 2.95     | 4.66|
| Mel-VAE-df4     | 3.10     | 4.74|
| Mel-VAE-df16     | 2.42     | 4.35|

**Text-to-code language model performances**: Next, we trained identical latent language models to generate codes for each of the Mel-VAEs and measured WER, CER, and speaker similarity. Similar to the trend in reconstructed audio quality, the below table shows that a higher code emitting frequency tends to increase speaker similarity. However, intelligibility, measured by WER and CER, performed better with a 16-fold compression compared to a 4-fold compression. This suggests that the longer the code length predicted, the more challenging it is for latent language models. Moreover, the default setting of an 8-fold compression shows the best intelligibility, indicating that our setting finds a sweet spot in balancing audio quality and latent language model prediction in the trade-off.
| Model | WER | CER | SIM-o |
| -------- | -------- | -------- | -------- |
| ByT5-base-df8 (default) | 2.79     | 1.00| 0.3879|
| ByT5-base-df4    | 4.56     | 2.32| 0.4117|
| ByT5-base-df16      | 3.20     | 1.19| 0.3629|

**Inference time**: We also reported the end-to-end inference time of 10s speech, varying with frequency.

| Model | Inference time |
| -------- | -------- |
| ByT5-base-df8 (default) | 3.46s |
| ByT5-base-df4    | 6.87s     |
| ByT5-base-df16      | 1.88s     |

We do believe this analysis not only strengthens our understanding of the system's performance under different conditions but also validates our choice of the default setting.

---

> ### Author Response · Authors · 2023-11-21
> **General Response to AC and Reviewer (cont.)**
>
> **Criticism 2: The performance of the proposed method compared to other methods**
>
> We are grateful for the insightful observations made by the reviewers (R2, R3) concerning the comparative performance of our proposed method against other methods. The reviewers pointed out that Voicebox outperforms our method, and noted that the incorporation of phoneme and duration information may not be a significant factor in explaining this performance gap.
>
> In light of these comments, we would like to clarify that our model can easily incorporate duration to improve the performance. We have validated and discussed several methods for performance enhancement in the subsequent paragraphs. Nonetheless, it is important to emphasize why our approach uniquely relies on plain text input, unlike other approaches. This is due to our model's ability to seamlessly integrate a pre-trained large language model, which is trained on a diverse array of texts and tasks. This integration allows us to achieve performance comparable to other baselines that employ more complex modeling.
>
> We incorporated the alignment search method from YourTTS to apply identified input durations and then evaluated its performance. The training involved adding the results of applying strided convolution to expanded character embeddings according to duration, to our decoder input embeddings. It is important to note that the only additional parameters in the resulting model are 164 character embeddings and a strided convolutional layer. The performance in terms of WER, CER, and speaker similarity is shown in the table below. When alignment from the alignment search was provided, we observed no quality gain in the continuation task.
>
> | Model | WER | CER | SIM-o | SIM-r |
> | ---------------- | ---- | ---- | ------ | ------ |
> | CLaM-en          | 2.36 | 0.79 | 0.4767 | 0.5128 |
> | CLaM-en (w/ duration) | 2.53 | 0.86 | 0.4876 | 0.5203 |
>
> However, we can observe a noticeable performance improvement in all metrics for the cross-sentence task, where our method exhibits greater degradation relative to other models and models are required to generate outputs from discontinuous texts and partially truncated audio prompts.
>
> | Model | WER | CER | SIM-o | SIM-r |
> | ---------------- | ---- | ---- | ------ | ------ |
> | CLaM-en          | 5.11 | 2.87 | 0.4951 | 0.5382 |
> | CLaM-en (w/ duration) | 3.59 | 1.55 | 0.5122 | 0.5484 |
>
> Note that even with only negligible additional parameters, significant performance enhancement occurs in our method when duration is provided, particularly in cross-sentence cases where the input text is concatenated with partially truncated sentences. This also explains the limitations of our model's performance in the cross-sentence task where continuous sentences are not provided.
>
> One potential solution to address this issue, akin to the approach used by SPEAR-TTS which maintains consistent performance in cross-sentence tasks, could be to explicitly inform the discontinuity between the audio prompt and the decoder output by a prompt separator token. This simple yet auxiliary method might help alleviate the identified problem. We acknowledge an error in the CER score (2.21) for SPEAR-TTS in the cross-sentence task and have corrected it to 1.92 in our manuscript (Table 1).

---

### Meta-Review · Area_Chair_cSkF · 2023-12-05

**Metareview:**

## Scientific Claims and Findings
The paper proposes a new model that uses a generative text-based LLM and neural audio codec to perform large-scale, zero-shot text-to-speech. One tradeoff that has arisen in such models is that to achieve good audio quality, the codec must generate parallel streams of discrete tokens, but modeling these parallel streams with an LM is challenging. If the parallel streams are simply flattened, the sequence length that the LM must accurately model increases greatly. Alternatively, multi-step generation processes (such as proposed by Vall-E) are possible, but increase the complexity of the final model. In this work, a different approach is proposed. The codec, called Mel-VAE, learns a discrete representation of speech using probabilistic residual vector quantization, while the LM predicts continuous latent vectors that, when quantized using the Mel-VAE, match ground-truth tokens. This allows for efficient audio generation with just a single step of the LM. Empirical comparisons using both objective and subjective measures show that the proposed model outperforms a number of competing models on an English continuation task (the original version of the paper used YourTTS as the baseline, while a revised version adds Vall-E, SPEAR-TTS, and Voicebox), except for Voicebox, while enjoying faster inference than Voicebox. The proposed model is also shown to perform reasonably well on an English cross-sentence task and a multi-lingual continuation task. An ablation experiment shows that the Mel-VAE leads to better objective metrics (PESQ and ViSQOL) than standard residual vector quantization and Encodec.

## Strengths
- The probabilistic vector quantization approach that allows parallel generation of token streams with a single step of the underlying LM is a novel algorithmic contribution in this area, and it is promising enough that I expect this paper to be useful to other researchers.

## Weaknesses
- The abstract oversells the work. The statement "CLaM-TTS is better than or comparable to state-of-the-art zero-shot TTS baselines regarding naturalness, intelligibility, speaker similarity, and inference speed" is overly optimistic in light of the results in Tables 1 and 2 that show Voicebox outperforming CLaM-en in all dimensions where a comparison can be made except for inference time. It would be better to discuss the relative advantages and disadvantages of the different modeling approaches (CLaM-TTS and Voicebox are quite different from one another) and to point out that if one wants to use a more text-based approach (like CLaM-TTS, SPEAR-TTS, or Vall-E), the CLaM-TTS architecture has competitive performance.
- The codeword rate experiments and dataset scaling experiments should go into the supplemental material.
- Is there a typo at the bottom of page 5 under "Inference"? Shouldn't it read "select k from the distribution $p_{\theta}(k|x,C_{<t})$"?
- Is CLaM supposed to be an acronym? If so, the paper fails to define it.
- The paper appears to claim that the probabilistic RVQ framework it uses improves codebook utilization: "The core of our method lies in the probabilistic discrete representation learning, ensuring that all discrete latent codes participate in the training process. [in the Introduction]" However, this claim is not substantiated in the experiments, though there is a comparison to conventional RVQ in terms of objective speech quality measures.

**Justification For Why Not Higher Score:**

There are enough unaddressed weaknesses in the paper (enumerated above) that I am not comfortable scoring it higher.

**Justification For Why Not Lower Score:**

The core algorithmic idea is a really nice one, and a strong contribution to large-scale TTS models that work directly from text instead of doing more fine-grained modeling from phones.

---

### Decision · Program_Chairs · 2024-01-16

Accept (poster)